# Anti-Bacterial and Anti-Biofilm Activities of Anandamide against the Cariogenic *Streptococcus mutans*

**DOI:** 10.3390/ijms24076177

**Published:** 2023-03-24

**Authors:** Goldie Wolfson, Ronit Vogt Sionov, Reem Smoum, Maya Korem, Itzhack Polacheck, Doron Steinberg

**Affiliations:** 1Institute of Biomedical and Oral Research (IBOR), Faculty of Dental Medicine, The Hebrew University of Jerusalem, Jerusalem 9112102, Israel; 2Institute for Drug Research, School of Pharmacy, The Hebrew University of Jerusalem, Jerusalem 9112102, Israel; 3Department of Clinical Microbiology and Infectious Diseases, Hadassah-Hebrew University Medical Center, Jerusalem 9112001, Israel

**Keywords:** *S. mutans*, anandamide, anti-bacterial, anti-biofilm, endocannabinoids

## Abstract

*Streptococcus mutans* is a cariogenic bacterium in the oral cavity involved in plaque formation and dental caries. The endocannabinoid anandamide (AEA), a naturally occurring bioactive lipid, has been shown to have anti-bacterial and anti-biofilm activities against *Staphylococcus aureus*. We aimed here to study its effects on *S. mutans* viability, biofilm formation and extracellular polysaccharide substance (EPS) production. *S. mutans* were cultivated in the absence or presence of various concentrations of AEA, and the planktonic growth was followed by changes in optical density (OD) and colony-forming units (CFU). The resulting biofilms were examined by MTT metabolic assay, Crystal Violet (CV) staining, spinning disk confocal microscopy (SDCM) and high-resolution scanning electron microscopy (HR-SEM). The EPS production was determined by Congo Red and fluorescent dextran staining. Membrane potential and membrane permeability were determined by diethyloxacarbocyanine iodide (DiOC2(3)) and SYTO 9/propidium iodide (PI) staining, respectively, using flow cytometry. We observed that AEA was bactericidal to *S. mutans* at 12.5 µg/mL and prevented biofilm formation at the same concentration. AEA reduced the biofilm thickness and biomass with concomitant reduction in total EPS production, although there was a net increase in EPS per bacterium. Preformed biofilms were significantly affected at 50 µg/mL AEA. We further show that AEA increased the membrane permeability and induced membrane hyperpolarization of these bacteria. AEA caused *S. mutans* to become elongated at the minimum inhibitory concentration (MIC). Gene expression studies showed a significant increase in the cell division gene *ftsZ*. The concentrations of AEA needed for the anti-bacterial effects were below the cytotoxic concentration for normal Vero epithelial cells. Altogether, our data show that AEA has anti-bacterial and anti-biofilm activities against *S. mutans* and may have a potential role in preventing biofilms as a therapeutic measure.

## 1. Introduction

A biofilm is an architectural colony of microorganisms encased in a matrix of extracellular polymeric substances (EPS) that they themselves produce [1]. Biofilms can contain homogeneous or heterogeneous populations of bacteria and fungi, can be single or multilayered, and can adhere to biotic and abiotic surfaces, including medical devices [1]. They are typically pathogenic in nature and can have many implications for human health, often causing infections and diseases [2]. In fact, bacteria that mature into multicellular biofilms are a clinical challenge. These biofilms protect the bacteria from the influence of the immune system and from anti-microbial agents and are therefore difficult to treat. They have been found to cause more than 65% of all bacterial infections in humans, including oral diseases [2].

Biofilms provide an environment that is advantageous and beneficial for the microorganisms embedded within them. One advantageous property of biofilms is the communication system between the microbes called quorum sensing, which helps regulate the growth and development of the biofilm [3,4]. Furthermore, their compact structure and surrounding EPS make the biofilms hard to penetrate and destroy [5]. Therefore, biofilms may confer tolerance to antibiotics and even to the immune system as the immune cells cannot reach the bacteria/fungi surrounded by the EPS [6].

The oral microbiota contains a unique level of microbial diversity, with up to a thousand diverse species of microorganisms colonizing there either as planktonic cells or incorporated into biofilms [6]. Dental implants and dentures, in addition to the natural dentition of the mouth, provide conducive environments for biofilm growth [6]. The surface of teeth, the spaces between teeth and the gingival cavity can be colonized with bacteria, making them hard to reach, and the biofilms that grow there are even more difficult to remove [6]. A mature oral biofilm can produce proteolytic enzymes that can cause direct damage to the soft and hard tissues as well as interfere with host defense mechanisms [6]. Moreover, bacteria in the biofilm and the substances they produce greatly contribute to the development of caries, gingivitis and periodontitis [7].

Among many virulent microorganisms in the oral cavity, *Streptococcus mutans* is known to be one of the most predominant, contributing to dental diseases such as dental caries. *S. mutans* exhibits many properties that make it a successfully virulent bacterium. These include its ability to adhere to solid surfaces, produce acid and survive in acidic environments [1,8].

The clinical shift toward disease prevention rather than treatment is a common theme in oral medicine. This change in concept is accompanied by better techniques of prevention and treatment of dental diseases. One new approach being explored is the potential use of endocannabinoids [9]. The endocannabinoid system is an endogenous system comprising the endocannabinoid compounds and their receptors. The CB1 and CB2 cannabinoid receptors can be found throughout the body, with higher expression of CB1 in the CNS and of CB2 on immune cells [10]. The endocannabinoid system has been found to modulate many physiological processes, including metabolism, appetite, mood, learning, memory and pain, as well as the cardiovascular, gastrointestinal and immune systems [10]. Endocannabinoids are produced upon demand and affect the nervous system by regulating the secretion of neurotransmitters. Additionally, endocannabinoids were found to have anti-microbial effects. For example, N-arachidonoyl ethanolamine (anandamide, AEA) was found to have anti-microbial and anti-biofilm activities against methicillin-sensitive and methicillin-resistant *Staphylococcus aureus* [10,11,12,13]. Fornelos et al. [14] observed that AEA prevented the growth of *Streptococcus salivarius* and *Enterococcus faecalis* when applying 17–35 µg/mL (50–100 µM) AEA. Additionally, the endocannabinoids have been found to have anti-fungal activities [15]. However, studies are lacking concerning their activities against the oral bacterium *S. mutans*. The aim of this study was to investigate the anti-bacterial and anti-biofilm effects of AEA on cariogenic *S. mutans* and shed light on the action mechanism.

## 2. Results

### 2.1. Anandamide (AEA) Has a Bactericidal Effect on S. mutans

Initially, we measured the optical density (OD) of planktonic growing *S. mutans* that had been incubated with various concentrations of AEA (1.25–50 µg/mL) for 24 h. This assay showed that the minimum inhibitory concentration (MIC) of AEA on *S. mutans* was 12.5 µg/mL (Table 1 and Figure 1A; *p* < 0.05). Notably, AEA has also an anti-bacterial effect against other oral bacteria, including *S. sobrinus* and *S. sanguinis* (Table 1).

To study whether AEA had a bacteriostatic or bactericidal effect on these bacteria, the number of colony-forming units (CFUs) of *S. mutans* grown in the absence or presence of 25 µg/mL AEA were studied over time. After a 1 h incubation, the number of CFUs was similar to that of the control, but after 2 h there was a two-fold reduction in CFUs in the presence of AEA, which was further reduced after 4 and 8 h to 99.9% (Figure 1B; *p* < 0.05). These observations suggest that AEA has a bactericidal effect on these bacteria. Notably, the bacteria did not recover even after a 24 h incubation (Figure 1B; *p* < 0.05).

Since acid production is a virulence factor of *S. mutans* involved in caries formation, it was also important to study the effect of AEA on this parameter. To this end, the pH of the culture medium of *S. mutans* grown in the presence or absence of various concentrations of AEA was measured at different time points. At concentrations of 1.56–6.25 µg/mL AEA, the pH of the culture medium steadily decreased during the first 6 h to a pH around 5, with a similar pattern for the control samples (Figure 1C). However, at 12.5 µg/mL and higher concentrations of AEA, the pH remained neutral, varying between 6.5 and 7.5 for all the time points tested (Figure 1C), which might have been due to the lack of bacterial growth at these concentrations (Figure 1D). When looking at the kinetic growth curve of *S. mutans* (Figure 1D), there was a delay in the onset of the log growth phase for *S. mutans* at concentrations between 1.56 and 6.25 µg/mL AEA, while no growth was observed at concentrations of 12.5 µg/mL AEA and above (Figure 1D). Since the pH values observed in Figure 1C go along with the changes in bacterial growth (Figure 1D), it is likely that AEA does not directly prevent acid production but rather that the prevention of acid production is due to the reduced number of bacteria.

### 2.2. AEA Prevents Biofilm Formation by S. mutans

Next, we tested the effect of AEA on the biofilm growth and development of *S. mutans*. To this end, the resulting biofilms were subjected to biomass Crystal Violet (CV) staining, metabolic MTT assay and SYPRO Ruby biofilm protein matrix staining. All three assays showed significant reductions in biofilm biomass of 87.9 ± 0.03%, 94.6 ± 0.004% and 52.9 ± 7%, respectively, at 12.5 µg/mL AEA in comparison to control samples (Figure 2A–C; *p* < 0.05). The metabolic assay demonstrated a higher reduction in biofilm formation than the CV and SYPRO Ruby dyes that stain EPS as well as proteins of both live and dead bacteria. This observation suggests that most of the bacteria were dead or metabolically inactive at this concentration. In the concentration range of 1.56–6.25 µg/mL AEA, there was a slight increase in biofilm formation (Figure 2A–C). The MBIC was determined to be 12.5 µg/mL, which is similar to the MIC value. Thus, the anti-biofilm effect of AEA seems to be largely caused by its anti-bacterial effect.

### 2.3. Live/Dead (SYTO 9/PI) and Dextran Staining of Biofilms Formed in the Presence of AEA

The EPS matrix is one of the most advantageous attributes for biofilms, creating a slim-like layer that facilitates the adhesion of additional bacteria to the surface. To investigate the effect of AEA on EPS production, untreated and AEA-treated *S. mutans* biofilms were analyzed after live/dead and EPS staining by spinning disk confocal microscopy (SDCM). SYTO 9 stains live and dead bacteria by green fluorescence, PI stains dead bacteria and extracellular DNA (eDNA) by red fluorescence, while fluorescent-labelled dextran 10,000 binds to EPS and is presented here as blue fluorescence. The fluorescence intensities in different layers of the biofilms were analyzed by capturing optical cross sections at 2.5 µm intervals from the bottom of the biofilm to its top (Figure 3A).

There was a significant increase in SYTO 9 staining at 1.56 µg/mL AEA (Figure 3B), which supports the increased biofilm biomass at the lower AEA concentrations seen in Figure 2C. However, the SYTO 9 staining was strongly reduced by 92.9 ± 2% and 99.5 ± 2% at 12.5 and 25 µg/mL AEA, respectively (Figure 3B), which accords with our data described above showing a strong reduction in biofilm formation at these concentrations (Figure 2). The PI staining was concomitantly reduced at these concentrations (Figure 3C), which was due to the significant reduction in the total numbers of bacteria in the biofilms under these conditions. However, when plotting the ratio of PI/SYTO 9 that represents the relative number of dead bacteria in comparison to the total number of bacteria in the biofilm, we observed that this ratio increased at 12.5 µg/mL AEA and higher concentrations (Figure 3H), indicating the presence of more dead bacteria, which is in accordance with the reduced MTT metabolic activity observed under these conditions (Figure 2B). The EPS production as determined by the intensity of fluorescent Dextran 10,000 showed significant reductions of 67 ± 8% and 96 ± 3% at 12.5 and 25 µg/mL, respectively (Figure 3F,G), which goes along with the reduced numbers of bacteria. In order to determine whether there was an alteration in EPS production per bacterium, the fluorescence intensity of dextran was plotted alongside the PI and SYTO 9 results (Figure 3H,I). We observed that the ratio of dextran staining to SYTO 9 staining was significantly higher in the 12.5 µg/mL AEA-treated biofilms (Figure 3I) than in the control biofilms (Figure 3H), indicating that the amounts of EPS produced per bacterium in the biofilms formed in the presence of AEA were higher than those for the control bacteria. The reason for this increase might have been the release of EPS-producing enzymes upon AEA-induced cell death, which continued to produce EPS.

### 2.4. AEA Leads to Increased Dextran Binding to Bacteria

To further study the ratio of EPS production per bacterium, *S. mutans* were exposed to various concentrations of AEA for 2 h and then stained with fluorescent dextran, followed by flow cytometry. There was a dose-dependent increase in dextran staining with increasing concentrations of AEA, reaching a maximum at 25 µg/mL (Figure 4A,B). This observation goes along with the SDCM findings (Figure 3H–I). We cannot exclude the possibility that dextran also binds to other components of the bacterial cell surface besides EPS.

### 2.5. The Congo Red Assay Shows Reduced EPS Secretion following AEA Treatment

To study whether EPS secretion is affected by AEA during the initial stages of biofilm formation, *S. mutans* were exposed to 12.5, 25 and 50 µg/mL AEA or 0.5% ethanol (control) for 2 h and 4 h and then plated on Congo Red agar plates for overnight incubation. The EPS, indicated by black coloration around the colony, were quantified. The control showed clear black staining around the colony, while bacteria that had been treated with 12.5, 25 or 50 µg/mL AEA showed dose-dependent reductions in EPS (Figure 5A,B). A significant EPS reduction of 94 ± 3% was seen at 50 µg/mL AEA (Figure 5C). These results suggest that AEA affects the production and/or secretion of EPS already after a 2 h incubation, which might contribute to its anti-biofilm activity.

### 2.6. AEA Alters the Membrane Properties of S. mutans

To measure the effect on membrane permeability, *S. mutans* were grown in planktonic conditions and treated with different concentrations of AEA for 2 h. The bacteria were stained with the nucleic acid binding dyes SYTO 9 and propidium iodide (PI) or the cell-permeable dye calcein AM indicative of cell viability (Figure 6A–C). Their relative fluorescence intensities were measured by flow cytometry (Figure 6D,E). SYTO 9 and calcein AM can freely diffuse into bacteria, while PI can only enter bacteria when membranes are perforated. Both SYTO 9 and calcein AM will leak out of bacteria when membranes are perforated. The flow cytometry results show that there was a reduction in the SYTO 9 and calcein AM fluorescence intensities concomitant with an increase in PI fluorescence intensity upon treatment with 12.5, 25 and 50 µg/mL AEA (Figure 6A–E). When inspecting the dot plots of the SYTO 9 and PI staining, a bacterial sub-population of SYTO 9^low^PI^high^ could be discerned, which reached 16.37% and 52.47% for bacteria treated with 25 and 50 µg/mL AEA, respectively (Figure 6F). This suggests increased membrane leakage and membrane permeability at these AEA concentrations. Notably, two peaks were observed for both dyes at 25 and 50 µg/mL AEA, which accords with the observation that around 50% of the bacteria had died after 2 h (Figure 1B).

### 2.7. AEA Leads to an Accumulation of DAPI in S. mutans

Initially, we intended to study the effect of AEA on DNA content using DAPI, which stains DNA but not RNA and freely enters bacteria due to its neutral charge. The control and AEA (2 h)-treated bacteria were simultaneously stained with Nile Red, which emits red fluorescence when integrated into the bacterial membrane. Surprisingly, we observed that AEA led to an intracellular accumulation of DAPI (six- to seven-fold) that reached a maximum at 12.5 µg/mL (Figure 7A,B). Notably, two peaks were observed at 25 and 50 µg/mL AEA, which might reflect different DNA contents [11]. AEA caused a decrease in the fluorescence intensity of Nile Red up to the concentration of 25 µg/mL (Figure 7C,D).

Since DAPI has been shown to be pumped out by efflux pumps in *S. mutans* [16], the intracellular accumulation of DAPI following AEA treatment might be due to AEA-mediated inhibition of these efflux pumps in *S. mutans*. In order to distinguish between an effect of AEA on the efflux mechanism and one on the DNA content, *S. mutans* that had been treated with varying concentrations of AEA for 2 h were fixed in methanol prior to DAPI staining. The fixation prevents the activity of the efflux pumps, such that the DAPI staining under these conditions represents the DNA content. The flow cytometry showed two peaks of the DAPI-stained AEA (2 h)-treated bacteria that appeared at similar intensities to the control bacteria (Figure 7E), indicating that the accumulation of DAPI observed in the live AEA-treated bacteria was due to efflux inhibition. Notably, in the AEA-treated bacteria, there was a dose-dependent increase in a DAPI^low^ bacterial population reaching 37.1 ± 2.0% with 12.5 μg/mL AEA (Figure 7E), which represented dead bacteria. These results confirm that there is a dose-dependent bactericidal effect on *S. mutans*, as was observed when counting the CFUs (Figure 1B) and double-staining the bacteria with SYTO 9 and PI (Figure 6F).

### 2.8. AEA Induces Immediate Membrane Hyperpolarization in S. mutans

The effect of AEA on membrane potential was determined by exposing *S. mutans* to various concentrations of AEA and then incubating them with the potentiometric probe DiOC2(3) reagent for 30 min prior to flow cytometry. Green fluorescence indicates the amount of dye taken up by the bacteria, and red fluorescence intensity reflects the membrane potential. AEA caused a slight increase in the red fluorescence intensity in comparison to the green fluorescence at 12.5 µg/mL AEA, while a strong increase in the red fluorescence intensity in comparison to the green fluorescence was observed in bacteria treated with 25 and 50 µg/mL AEA (Figure 8A–C), indicating that AEA induces an immediate increase in the membrane potential. At the highest concentration of 50 µg/mL AEA, there was an increase in the uptake of DiOC2(3), which goes along with the increased DAPI uptake and might have been due to inhibition of an efflux pump, as discussed above.

### 2.9. AEA Affects the Cell Length and Morphology of S. mutans

To visualize the effect AEA has on cell morphology, *S. mutans* were grown in the absence or presence of different concentrations of AEA, and the samples were analyzed by high-resolution scanning electron microscopy (HR-SEM). For planktonic growing bacterial samples, the bacteria were exposed to AEA for 2 h (Figure 9A,B), while for the biofilm samples, the bacteria were incubated with AEA for 24 h (Figure 9D–F). The planktonic growing bacteria that had been treated with 12.5 µg/mL AEA for 2 h appeared to be dysmorphic, swollen and elongated, as indicated by the white arrows (Figure 9B), in comparison to the control bacteria (Figure 9A). The AEA-treated samples showed an abundance of several rings along the bacteria (Figure 9B, white arrows). After measuring the lengths of the samples, the control bacteria were found to be on average 0.75 ± 0.14 µm, while the samples treated with 12.5 µg/mL AEA were found to be on average 0.88 ± 0.22 µm (Figure 9C). The bacteria in the biofilms formed in the presence of 25 or 50 µg/mL AEA also appeared longer than the control bacteria and with many disrupted membranes (Figure 9E,F) as compared to the control samples (Figure 9D). The average length of the control samples was measured to be 0.74 ± 0.16 µm, while the lengths of the samples treated with 12.5, 25 and 50 µg/mL AEA, on average, were 0.94 ± 0.19 µm, 0.96 ± 0.19 µm and 0.85 ± 0.16 µm, respectively (Figure 9G).

### 2.10. AEA Affects the Gene Expression of the Cell Division Gene ftsZ in S. mutans

To study the molecular basis of the anti-bacterial and anti-biofilm activities of AEA in *S. mutans*, changes in gene expression were determined in bacteria exposed to 12.5 μg/mL AEA for 4 h in BHIS by real-time qPCR. The study focused on genes related to biofilm formation (Figure 10A), cell division (Figure 10B), antioxidant defenses (Figure 10B), stress and stringent responses (Figure 10B), quorum sensing (Figure 10B), and acid tolerance (Figure 10C). AEA had only minor effects on the expression of genes associated with biofilm formation, adhesion and EPS production, except for the three-fold upregulation of the beta glycosyltransferase *gtfB* involved in EPS production, the two-fold increase in the surface protein antigen *spaP* involved in adhesion, and the two-fold increase in the Sec translocase *secA* involved in the secretion of virulence factors (Figure 10A). Outstanding was the 4.2–8.2-fold induction of the cell division gene *ftsZ* involved in FtsZ-ring formation during cell division (Figure 10B), which might explain the growth-inhibitory action of AEA. There were no significant alterations in the genes associated with antioxidant defense (*nox* and *sodA*) (Figure 10B), stress response (*dnaK* and *groEL*) and stringent response (*relA*) (Figure 10B), nor were there any significant changes in the tested quorum-sensing genes (*vicR* and *luxS*) (Figure 10B) or the genes associated with acid stress (*pdhA*, *atpD*, *aguD*, *fabM* and *glgP*).

### 2.11. AEA Causes a Reduction in Preformed Biofilms

In order to analyze the effect of AEA on preformed biofilms, *S. mutans* were grown in BHIS for 4 h to form biofilms prior to treatment with various concentrations of AEA for 24 h. The metabolic activity of the resulting biofilms was measured using the MTT assay. There was a significant increase in the biofilm metabolic activity at concentrations of 1.56–6.25 µg/mL AEA (Figure 11), which resembled the effect previously observed for AEA during biofilm formation (Figure 2C and Figure 3). No significant effect was observed with 12.5 and 25 mg/mL AEA, while a significant reduction in metabolic activity of 56 ± 0.4 % was observed with 50 µg/mL AEA (Figure 11).

### 2.12. The Anti-Bacterial Concentrations of AEA Were Below Those Cytotoxic to Vero Epithelial Cells

To investigate the cytotoxicity levels of AEA, Vero kidney epithelial cells were exposed to various concentrations of AEA for 24 h, followed by MTT metabolic assay and visual inspection of the cells by light microscopy. There was no significant effect of AEA on the metabolic activity or morphology of Vero cells up to a concentration of 50 µg/mL (Figure 12). However, increasing the AEA concentration to 100 µg/mL induced cell death (Figure 12). Thus, the MIC of 12.5 µg/mL for AEA against *S. mutans* is below the cytotoxic value for Vero cells.

## 3. Discussion

Oral biofilms have been linked to several infections, including caries, periodontitis and gingivitis. Furthermore, dentures and implants can serve as substrates for biofilms to grow on and cause infections [6,7]. *Streptococcus mutans*, in particular, is one of the main etiological factors in dental caries [1]. Its virulence can be attributed to its ability to adhere to and form biofilms on teeth, produce acid, and tolerate acidic conditions [17]. With dental caries being among the most common infectious diseases in the oral cavity and one of the most common childhood illnesses [17], new therapies to treat dental caries are urged.

Targets for treating dental caries include reducing bacteria growth, strengthening the teeth, enhancing saliva production and decreasing the consumption of sugar in the diet [18]. Despite the abundance of approaches being researched, there are no Food and Drug Administration (FDA)-approved drugs for the actual treatment of dental caries. Rather, the most widely used FDA-approved product is a preventative measure, namely, fluoridated toothpaste, among other fluoride-containing products [18].

Aside from the conventional anti-caries therapies, such as fluoride and chlorhexidine, which are characterized by undesired side-effects [19], alternative natural compounds including cannabinoids, are emerging as novel and effective treatments against *S. mutans*. A range of studies performed on several cannabinoids, including cannabigerol (CBG) and cannabidiol (CBD), have revealed anti-bacterial as well as anti-biofilm activities against oral biofilms [20,21,22]. In addition to cannabinoids, another alternative treatment being studied is the use of endocannabinoids; albeit their dissimilar structure to cannabinoids, they are also natural compounds which have been found to have anti-microbial and anti-biofilm activities, in addition to anti-inflammatory and neuromodulatory activities [10]. However, there is a lack of research when it comes to the microbial targets of endocannabinoids. In recent years, the endocannabinoid anandamide (AEA) has been tested alone and in combination with other anti-microbial agents in relation to the growth of different microbes. In Fornelos et al. [14], among the strains tested, the most susceptible bacteria to AEA included *Streptococcus salivarius, Bacteroides fragilis* and *Enterococcus faecalis*. AEA was also found to act on multidrug-resistant bacterium *Staphylococcus aureus* (MDRSA) [11], suggesting that its mechanism of action is unaffected by drug-resistant mechanisms. This discovery could bring new ideas to light for drug-resistant illnesses and infections. For these reasons, our research focused on AEA and how it affects *S. mutans*.

In the present study, we observed that AEA had an MIC and MBIC of 12.5 µg/mL towards *S. mutans* and a transient bacteriostatic effect at 1.25–6.25 µg/mL. By counting the colony-forming units, AEA was shown to have a bactericidal effect on *S. mutans* at 12.5 µg/mL, which contrasts with its action on *Staphylococcus aureus* strains, where the effect was bacteriostatic with recovery of the bacteria at later time points [11]. The bactericidal effect seems to be due to increased membrane permeability, as shown by an increase in the SYTO 9^low^PI^high^ bacterial population, even after a 2 h incubation with AEA. Another support for the bactericidal effect can be drawn from Calcein AM staining, which is used to determine cell viability. We observed a dose-dependent reduction in Calcein AM staining with increasing concentrations of AEA, even after a 2 h incubation. Moreover, at the higher concentration of 50 µg/mL AEA, a Calcein AM^low^ population appeared which reflects dead bacteria. This population was also sometimes observed after a 2 h incubation with 25 µg/mL AEA. Another striking finding is the immediate membrane hyperpolarization of *S. mutans* caused by AEA. Membrane hyperpolarization of *S. mutans* has been associated with anti-bacterial actions of several compounds, including CBG [20], CBD [22] and epigallocatechin gallate (EGCG) [23]. Interestingly, AEA and CBD have been shown to cause membrane depolarization in *S. aureus* species [11,24], suggesting different regulations of membrane potential in these bacterial species. In *S. mutans*, membrane potential is regulated by the F_1_F_0_-ATPase [25], which is also involved in acid tolerance by being required for proton pump activity [26]. In this context, it is worth mentioning that AEA has been shown to reduce membrane ATPase activity in *S. aureus* [10]. It is therefore likely that one of the effects of AEA on *S. mutans* might be the targeting of F_1_F_0_-ATPase. Further studies are required to find the molecular targets of AEA.

Moreover, we observed that AEA caused intracellular accumulation of the nucleic acid dye DAPI. This is interesting in light of previous findings showing that AEA causes accumulation of DAPI in multidrug-resistant *S. aureus* [11], and this accumulation was related to inhibition of efflux pump activity, resulting in the sensitization of these bacteria to various antibiotics [11,27]. So far, only a few efflux pumps have been characterized in *S. mutans* [16,28,29,30], and some ATP-driven membrane secretory transport systems have been described (e.g., SecA) which are involved in the secretion of virulence factors, such as Pac (AgI/II), Gtf and Ftf [31]. In this context, it is notable that *secA* was upregulated by AEA, which might be a feedback mechanism to overcome the inhibition. Among the efflux pumps described in *S. mutans* are ATP-binding cassette (ABC) transporters which function as multiple sugar metabolism transporters [32]. Such transporters are promising targets for anti-microbial strategies [33].

It is likely that AEA has a general inhibitory effect on various efflux mechanisms, and this might explain the reduced EPS production observed by Congo Red staining, which, in turn, contributes to diminished biofilm formation. The discrepancies between the Congo Red and dextran stainings can be explained by their different binding partners. Congo Red turns black when it comes into contact with EPS, while the anionic dextran interacts also with positively charged components besides EPS. Notably, the dextran binding to the bacteria was enhanced following AEA treatment, which might explain, in part, the relatively higher dextran staining of the biofilms formed in the presence of AEA as detected by SDCM, despite the fact that most of the bacteria were dead.

When studying the DNA contents of bacteria after fixation, which eliminates the influence of the efflux pumps, we observed the appearance of two bacterial populations with different DAPI fluorescence intensities that represent different cell division stages (1N and 2N). These two peaks were also seen in the AEA (2h)-treated bacteria, although a dose-dependent DAPI^low^ population appeared, which reflects cells that have lost their DNA contents and are thus dead bacteria. Thus, the response of *S. mutans* to AEA differs from that of *S. aureus*, where AEA inhibits the division of the bacteria at the stage of daughter cell separation [11] as a result of altered autolysin activity [10]. Both *S. mutans* and *S. aureus* responded to AEA by becoming enlarged [11], but the mechanisms seem to differ. In *S. mutans*, the bacteria became swollen and were on their way to dying, while in *S. aureus* the enlarged cells had defective septa, leading to their arrest at the last stage of cell division. The swelling of *S. mutans* in response to AEA might be a result of osmotic stress due to altered membrane permeability and membrane potential.

Gene expression studies showed a significant induction of *ftsZ* expression by AEA, which might explain the growth-inhibitory effect. The cell division is tightly regulated, and expression of *ftsZ* that is too high or too low can interfere with proper cell division [34,35,36]. The increase in *ftsZ* gene expression might also be a compensation mechanism for the growth-inhibitory effect of AEA.

The anti-biofilm effect of AEA on *S. mutans* seems to go along with its anti-bacterial effect. However, the reduction in EPS production observed following AEA treatment suggests that AEA might also have a direct anti-biofilm activity. The anti-biofilm effect of AEA was associated with reduced biofilm biomass, less proteins and EPS, and significantly diminished metabolic activity. A specific anti-biofilm activity has previously been shown against *S. aureus*, where the MBIC was eight-fold lower than the MIC and was related to impaired slime production [12]. By using Congo Red agar plates, AEA was found to prevent EPS production in *S. mutans* in a way that strikingly resembles our previous findings for *S. aureus* [10]. As EPS serves as a binding agent for bacterial adhesion, its inhibition by AEA might contribute to biofilm prevention.

AEA was found to be more efficient in preventing biofilm formation than acting on preformed biofilms, although a certain reduction in preformed biofilm could be observed with the higher concentration of 50 µg/mL AEA. This is actually not surprising, as AEA is a hydrophobic molecule, while the EPS of the biofilm is relatively hydrophilic and thus expected to act as a drug barrier. Further studies are required to find the drug combination that can overcome this drug barrier, thus enabling the action of AEA and other anti-bacterial drugs on mature biofilms.

Additionally, sub-MBIC concentrations of AEA increased the EPS and biofilm biomass in comparison to control bacteria. Such a pro-biofilm effect has also been shown for other anti-microbial compounds at sub-MIC concentrations. For instance, Dong et al. [37] observed that at sub-MIC concentrations of NaF and chlorhexidine (CHX), *S. mutans* biofilms became denser with thicker EPS. One mechanism for the increased biofilm mass might be through the release of extracellular DNA and EPS-producing enzymes by a small bacterial population that has undergone cell death.

To achieve a better understanding of the anti-biofilm effect of AEA, we performed gene expression analysis of genes related to biofilm formation. Surprisingly, there was no significant downregulation of the tested genes in the AEA-treated bacteria, and the three genes *gtfB* (involved in EPS production), *spaP* (a surface protein adhesion molecule) and *secA* (involved in the secretion of virulence factors) were even upregulated. The other biofilm-related genes, as well as the quorum-sensing genes *vicR* and *luxS*, were not significantly altered by AEA. Thus, the anti-biofilm action of AEA seems to depend on a post-translational mechanism.

It was also important to study the cytotoxicity levels of AEA on normal Vero epithelial cells. AEA had no observable cytotoxic effect on Vero cells at the concentrations required for anti-bacterial and anti-biofilm effects against *S. mutans*, and the Vero cells even showed intact metabolic activity at 50 µg/mL. The cytotoxicity level of AEA on Vero cells was determined to be 100 µg/mL, which is four times the MIC/MBIC concentration against *S. mutans*, leaving a sufficient therapeutic window. It is important to mention that AEA is a naturally occurring compound produced by the human body. AEA was found to have anti-inflammatory and anti-nociceptive properties when administrated to mice [38,39,40]. Anandamide has also anxiolytic and neuroprotective activities [41]. These findings suggest that AEA has potential therapeutic applications. The combined anti-microbial and anti-inflammatory properties of AEA might have clinical relevance in caries, gingivitis and periodontitis.

It should be emphasized that the anti-biofilm activity of AEA was tested in a static biofilm model, and further studies are required to analyze its effects in a flow cell model that better mimics the continuous saliva flow in the oral cavity. Nevertheless, it was important to determine whether AEA has an anti-biofilm effect on *S. mutans* before proceeding with a flow cell model. Indeed, we observed that AEA prevents biofilm formation by *S. mutans*, although it is less effective against preformed biofilms. The data obtained in this study provide a basis for further studies aiming to find the optimal drug combination that can combat mature biofilms. A limitation of our study is that we mainly tested the anti-bacterial effect of AEA on *S. mutans*, even though the oral cavity is inhabited by hundreds of different microbial species. However, some initial anti-bacterial indications were presented for other oral bacteria tested, including *S. sobrinus* and *S. sanguinis*. Our study chose to focus on *S. mutans* since this oral bacterium is considered a major cause of caries. Now that we have established the effects of AEA on this bacterium, future experiments should focus on heterogeneous biofilms containing various common oral bacteria and fungi.

In summary, the present research demonstrates an anti-bacterial and anti-biofilm effects of AEA against *S. mutans.* One of these mechanisms seems to be the alteration of membrane potential, membrane permeability and membrane transport, which ultimately cause morphological changes in and the swelling of the bacteria, resulting in cell death. We believe that AEA is a potential drug in the preventative treatment of dental caries, and further work is needed to establish its efficacy and safety in vivo.

## 4. Materials and Methods

### 4.1. Materials

Anandamide (AEA) (>98.0% purity) was purchased from Cayman Chemical (Ann Arbor, MI, USA) and dissolved in absolute ethanol at a concentration of 10 mg/mL. Respective dilutions of ethanol (0.0156–0.5%) as well as untreated bacteria were used as controls in this study.

### 4.2. Bacterial Growth and Biofilm Formation

*S. mutans* UA159, *Streptococcus sanguinis* ATCC 10556 and *Streptococcus sobrinus* ATCC 27351 were grown overnight at 37 °C in 95% air/5% CO_2_ in brain heart infusion broth (BHI, Acumedia, Lansing, MI, USA) until an OD_600nm_ of approximately 1.2–1.3 was reached [20]. For planktonic growth, the overnight culture was diluted to an OD_600nm_ of 0.1 in BHI, while for biofilm formation the overnight culture was diluted to an OD_600nm_ of 0.1 in BHI containing 2% sucrose (BHIS). Two hundred microliters of the bacterial culture were seeded in each well of a tissue-grade flat-bottom 96-well microplate (Corning, Glendale, AZ, USA) in the absence or presence of increasing concentrations of AEA (1.25–50 µg/mL) or respective ethanol concentrations (0.0156–0.5%) and incubated at 37 °C in 95% air/5% CO_2_ for 24 h. At the end of incubation, the medium was removed, and the biofilms formed at the bottom of the wells were washed twice with 200 µL phosphate-buffered saline (PBS) to remove any remaining planktonic bacteria. Untreated and ethanol-treated bacteria served as controls. To study the effect of AEA on preformed biofilms, *S. mutans* were cultivated in BHIS for 4 h to allow biofilm formation, and the formed biofilms were washed twice in PBS before being incubated for 24 h with increasing concentrations of AEA or respective ethanol concentrations.

### 4.3. Colony-Forming Units (CFUs)

The numbers of bacteria in the untreated and treated samples were determined by ten-fold serial dilutions in BHI and then seeding 100 µL of each dilution onto BHI agar plates, followed by an overnight incubation at 37 °C in the presence of 95% air/5% CO_2_. After incubation, images were capture by Gel Imager–Fusion FX (Vilber Lourmat, Marne-la-Vallee, France), and ImageJ software was used to count the number of colonies on each plate. The following equation was used to calculate the CFUs per well in the original sample: number of colonies × dilution factor × original volume of sample × 10 [42].

### 4.4. Kinetic Analysis of Planktonic Growing Bacteria and Simultaneous pH Measurements

Bacterial suspensions of OD_600nm_ = 0.1 in BHI medium were treated with different concentrations of AEA (1.25–50 μg/mL) and incubated at 37 °C, air/5% CO_2_ for 24 h. At various time points, the pH of the samples was measured using pH indicator paper strips (MColorpHast, Merck KGaA, Darmstadt, Germany), and the optical density at 595 nm was measured in parallel in a Tecan Infinite M200 PRO plate reader (Tecan Trading AG, Männedorf, Switzerland) [20].

### 4.5. Crystal Violet (CV) Staining of Biofilms

The biofilms were stained with 200 µL of 0.1% Crystal Violet (CV) that was prepared from a 0.4% Gram’s crystal violet solution (Merck, EMD Millipore Corporation, Billerica, MA, USA) diluted with DDW [12]. After a 15 min incubation at room temperature, the CV solution was removed, and the wells were washed twice with DDW and dried. Extraction of the CV stain was performed by adding 150 µL of 33% acetic acid to the wells, followed by a 5 min incubation under constant shaking. The absorbance was measured at 595 nm using the M200 Tecan plate reader, which provides a measure of the amount of biofilm biomass.

### 4.6. MTT Metabolic Assay

This colorimetric assay is useful for measuring the metabolic activity of bacteria. The 3-(4,5-Dimethyl-2-thiazolyl)-2,5-diphenyl-2H-tetrazolium bromide (MTT) (Sigma Aldrich, St. Louis, MO, USA) assay was performed as previously described [21]. Briefly, 50 µL of a 0.5 mg/mL MTT solution in PBS was added to the biofilms in 96-well plates. After 1 h incubation at 37 °C, the wells were washed with 150 µL PBS, and the tetrazolium precipitates contained in the biofilms were extracted with 150 µL dimethylsulfoxide (DMSO) (Bio-Lab Ltd., Jerusalem, Israel). After 10 min on an orbital shaker, the absorbance was measured at 570 nm using the M200 Tecan plate reader.

### 4.7. FilmTracer SYPRO Ruby Biofilm Matrix Staining

SYPRO Ruby stain labels most classes of proteins and is used to stain proteins in the extracellular matrices of bacterial biofilms. The washed biofilms were stained with 100 μL of the FilmTracer SYPRO Ruby biofilm matrix stain (Invitrogen, Molecular Probes, Eugene, OR, USA) for 30 min at room temperature, followed by several washes with DDW [42]. The fluorescence intensity of the stained biofilms was measured in the M200 Tecan plate reader with an excitation at 450 nm and an emission at 610 nm.

### 4.8. Biofilm Analysis by Spinning Disk Confocal Microscopy (SDCM)

SDCM was used to examine the structure of the biofilms after treatment with AEA and to detect the presence of live/dead bacteria and extracellular polysaccharides (EPS). The biofilms were grown in the absence or presence of various concentrations of AEA in 12-well tissue culture plates for 24 h and then washed twice with PBS. Thereafter, the biofilms were stained with 3.3 µM SYTO 9 (Molecular Probes, Life Technologies, Carlsbad, CA, USA), 10 µg/mL of propidium iodide (PI) (Sigma, St. Louis, MO, USA) and 10 µg/mL of Alexafluor^647^-conjugated dextran 10,000 (Invitrogen, Thermo Fisher Scientific, Eugene, OR, USA) for 20 min at room temperature [22]. The SYTO 9 green fluorescence dye, which enters both live and dead bacteria, was visualized using 488 nm excitation and 515 nm emission filters. The PI red fluorescence dye, which only penetrates dead bacteria, was measured using 543 nm excitation and 570 nm emission filters. Thus, live bacteria fluoresce green light, while dead bacteria fluoresce both green and red light. The samples were visualized for thickness and bacterial vitality using the Nikon Yokogawa W1 Spinning Disk Confocal Microscope (Nikon Corporation, Tokyo, Japan) with 50 μm pinholes. The biofilm depth was assessed by capturing optical cross sections at 2.5 μm intervals from the bottom of the biofilm to its top. Three-dimensional images of the formed biofilms were constructed using the NIS-Element AR software. This software was also used to analyze the fluorescence intensities of SYTO 9, PI and fluorescent dextran staining in each of the captured layers of the biofilms. For biofilms treated with AEA, the sums of fluorescence intensities for all layers of the biofilm were presented and compared with those of the untreated control [42].

### 4.9. Determination of Extracellular Polysaccharide (EPS) Production by Congo Red Assay

The Congo Red agar method was used to determine the EPS production of *S. mutans* that had been incubated for 2 h and 4 h in the absence and presence of 12.5, 25 and 50 µg/mL AEA. BHI agar plates containing Congo Red and 1% sucrose were divided into quadrants in which 10 µL of the control and AEA-treated *S. mutans* cultures were seeded, followed by a 24 h incubation at 37 °C in 95% air/5% CO_2_. The black color that appears around the biofilm represents the EPS produced. The Image J program (The National Institute of Health, Bethesda, MD, USA) was used to analyze the area of the black coloration. The area of the EPS was calculated by subtracting the area of the bacterial colony from the total black area. The Congo Red agar plates were prepared by diluting 10 mL of an autoclaved 0.8% stock solution of Congo Red stain in 100 mL of autoclaved BHI containing 1.5% agar to which sucrose was added to a final concentration of 1% [21].

### 4.10. Determination of EPS Production of Individual S. mutans Cells Using Flow Cytometry

Overnight cultures of S. mutans were diluted in fresh BHI media to an OD of 0.2. Then, the cells were incubated in the absence or presence of various concentrations of AEA (1.56–50 µg/mL) and 0.05% ethanol for 2 h in fresh BHI media at 37 °C in 95% air/5% CO_2_. After incubation, the samples were stained with 10 µg/mL Alexafluor^647^ -conjugated anionic Dextran 10,000 (Invitrogen Live Technologies Corporation, Eugene, OR, USA) for 20 min at 37 °C in 95% air/5% CO_2_. Fluorescence intensity was analyzed by flow cytometry (LRS-Fortessa flow cytometer, BD Biosciences, San Jose, CA, USA) using the 640 nm laser excitation, and the fluorescence was collected using the 670 nm filter [13].

### 4.11. Membrane Permeability Assay

Control or AEA-treated bacteria were exposed to 10 µg/mL propidium iodide (PI) (Sigma) together with 3.3 µM SYTO 9 or 10 µM calcein AM staining (Biolegend, San Diego, CA, USA) for 20 min at 37 °C in order to assess membrane permeability and viability, respectively. PI enters only those cells whose membranes have been compromised and fluoresces in the red spectrum when binding to nucleic acids within the cells [43]. SYTO 9 stains live and dead bacteria and fluoresces green when binding to the nucleic acids within cells. Calcein AM diffuses passively into the cytoplasm, where it is converted into green-fluorescent calcein by native esterases. Calcein is retained in live cells but will leak out of cells whose plasma membranes have been perforated. The 488 nm laser excitation was used for PI and SYTO 9. The data were collected using the red and green filters, respectively [20].

### 4.12. DAPI and Nile Red Staining

Control or AEA-treated bacteria were exposed to 1 µg/mL of diamidino-2-phenylindole (DAPI) for 30 min together with 10 µg/mL Nile Red (APExBIO, Boston, MA, USA) for 30 min at 37 °C. After incubation, the samples were analyzed by flow cytometry (LRS-Fortessa flow cytometer, BD Biosciences). The 355 nm laser was used for DAPI, and the fluorescence was collected by the blue (450 nm) filter. DAPI is a blue-fluorescent probe that fluoresces brightly upon selectively binding to DNA [11]. The Nile Red was analyzed using the 561 nm yellow–green laser excitation, and data were collected using the 635 nm filter [20].

### 4.13. Measuring DNA Content with DAPI Staining

An overnight bacterial culture was suspended in an OD_600nm_ = 0.1 in BHI medium for 2 h incubation in order to allow the bacteria to reach the log phase. After this incubation, the bacteria were resuspended at an initial OD_600nm_ of 0.3 and treated with different concentrations of AEA (1.25–50 µg/mL), and then incubated for another 2 h at 37 °C, air/5% CO_2_. Thereafter, the bacteria were centrifuged and resuspended in 50 µL PBS to which 500 µL methanol was added drop-wise by vortexing the tube. The samples were then incubated in methanol overnight at −20 °C, after which the bacteria were centrifuged again and washed and rehydrated with PBS. Then, DAPI was added at a final concentration of 1 µg/mL to the samples and allowed to incubate at room temperature for 30 min. The cells were analyzed using the 355 nm excitation laser for DAPI, and the fluorescence was collected by the 450 nm filter [11].

### 4.14. Membrane Potential Assay

The immediate effect of AEA on the membrane potential of *S. mutans* was analyzed using the BacLight Membrane Potential Kit (Molecular Probes, Life Technologies, Eugene, OR, USA), according to the manufacturer’s instructions. An overnight culture of *S. mutans* was resuspended in PBS at an OD_600nm_ of 0.3 and exposed to various concentrations of AEA. Immediately thereafter, 3,3′-diethyloxacarbocyanine iodide (DiOC2(3)) was added to a final concentration of 30 μM. After 30 min, the bacteria were analyzed by flow cytometry (LSR-Fortessa flow cytometer, BD Biosciences) using the 488 nm excitation laser, and the data were collected using the green (530 nm) and red (610/620 nm) filters [22]. The BD FACSDiva software was used for the collection of data, and the FCS Express 7 software was used to analyze the data [22].

### 4.15. High-Resolution Scanning Electron Microscopy (HR-SEM)

*S. mutans* bacteria were grown in planktonic conditions with BHI media and in BHIS to analyze biofilm formation in the absence or presence of various concentrations of AEA. The biofilms were allowed to form on sterile square glass pieces. The bacteria in planktonic conditions were incubated for 2 h, while the biofilms were allowed to form for 24 h. After the incubation period, the glass specimens were rinsed with DDW and fixed in 4% glutaraldehyde in DDW for 40 min. The glass specimens were washed again with DDW and allowed to dry at room temperature. The specimens were then mounted on a metal stub and sputter-coated with iridium and visualized by a high-resolution scanning electron microscope (Magellan XHR 400 L, FEI Company, Hillsboro, OR) [20]. Three specimens from each treatment group were prepared and examined under HR-SEM to evaluate the effect of AEA on planktonic and biofilm formation. ImageJ software was used to measure the length of each bacterial cell in the images.

### 4.16. RNA Extraction

RNA extraction was performed as previously described in Sionov et al. [13], with slight modifications. An overnight culture of *S. mutans* was diluted to an OD of 0.1 in BHI containing 2% sucrose, and the bacteria were incubated for 4 h in the absence or presence of 12.5 µg/mL AEA or 0.125% ethanol. At the end of incubation, the bacterial pellet was incubated in 1 mL of RNA protect (Qiagen, Hilden, Germany) for 5 min at room temperature. Thereafter, the bacterial pellet was resuspended in 1 mL of Tri-Reagent (Sigma-Aldrich) and transferred to Type B bead tubes (Macherey-Nagel, Düren, Germany). After 3 disruptions for 45 s at a speed of 4.5 m/sec using a FastPrep cell disrupter (BIO 101, Savant Instruments, Holbrook, NY, USA), with 5 min breaks on ice in between each cycle, the glass beads were removed by centrifugation (2 min at 20,000×g). Thereafter, 200 µL of chloroform was added, followed by hard vortexing for at least 15 sec. After a 15 min incubation at room temperature, when the two phases have become separated, the samples were centrifuged for 15 min at 21,100×g at 4 °C. The upper water phase containing the RNA was collected and mixed with an equal volume of isopropanol, followed by a 30 min incubation at room temperature before being centrifuged for 30 min at 21,100× *g* to precipitate the RNA. The RNA pellet was washed twice with 1 mL of 75% EtOH and allowed to dry for 30 min before being resuspended in RNase- and DNase-free water (Bio-Lab., Jerusalem, Israel). RNA purity was determined by running the samples in a 1% agarose gel containing ethidium bromide, and the quantity was determined by a Nanodrop (Nanovue, GE Healthcare Life Sciences, Buckinghamshire, UK).

### 4.17. Reverse Transcription (RT) and Quantitative Real-Time PCR

The AB high-capacity cDNA reverse transcription kit (Applied Biosciences by Thermo Fisher Scientific, Vilnius, Lithuania) was used to transcribe RNA to cDNA. The relative expression levels of target genes were analyzed by the Bio-Rad CFX Connect Real-time system with the Bio-Rad CFX Maestro program. Power SYBR Green PCR Master mix (Applied Biosystems, Life Technologies, Woolston Warrington, UK) was used to amplify the genes of 10 ng cDNA per well in combination with 300 nM of respective F/R primer set (Table 2). The PCR cycle involved an initial heating at 50 °C for 2 min followed by an activation step at 95 °C for 10 min and then 40 cycles of amplification (95 °C for 15 s and 60 °C for 1 min). The dissociation curve was determined by initial heating at 95 °C for 15 s, followed by 10 s at 60 °C, with 0.5 temperature increments until 95 °C was reached. *gyrA* served as the housekeeping gene [44] for calculating the relative changes in target gene expression by the 2^−ΔΔCt^ method. The fold change of each gene for each treated sample was analyzed against each control sample, and the averages of the obtained values are presented. Gene expression is expressed in relative values, with the expression level of the control sample set to 1 for each gene. The assays were performed in triplicate and repeated three times [13].

### 4.18. Cytotoxicity Assay using Vero Kidney Epithelial Cells

Vero cells were seeded in 200 µL of fresh Dulbecco’s Modified Eagle’s Medium (DMEM) supplemented with 10% Fetal calf serum (FCS) at a density of 5 × 10^4^ cells per well of a 96 flat-bottom tissue culture well plate (Corning) and incubated overnight in a humidified incubator at 37 °C in the presence of 5% CO_2_. The following day, the medium above the confluent monolayer was removed and new medium containing various concentrations of AEA or an equal dilution of ethanol was added, followed by an incubation for another 24 h. During the last 30 min of incubation, MTT was added to the medium at a final concentration of 1 mg/mL, and the metabolic activity was determined by reading the absorbance at 570 nm. The morphology of the cells was inspected under a light microscope to visualize any cytotoxic effects before adding the MTT [22].

### 4.19. Statistical Analysis

The experiments were performed independently three times in triplicate, and the data were analyzed statistically using the Student’s *t*-test in Microsoft Excel, with a *p*-value of less than 0.05 considered significant when comparing treated versus control samples.

## Figures and Tables

**Figure 1 ijms-24-06177-f001:**
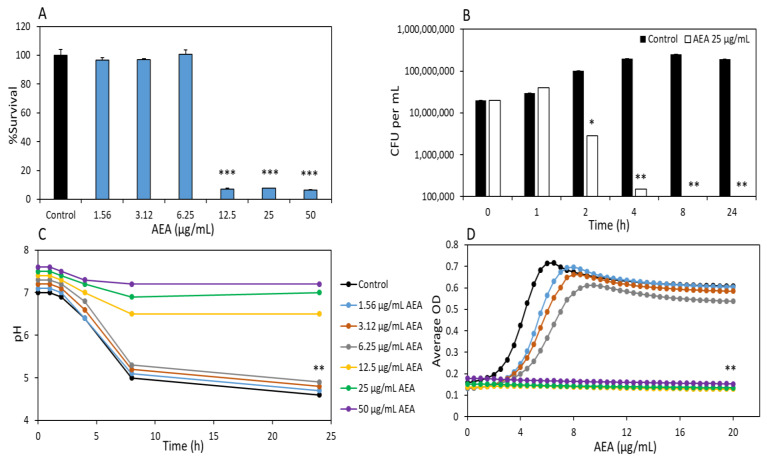
AEA prevents bacterial growth and has a bactericidal effect on *S. mutans.* (**A**) The viability of *S. mutans* after a 24 h incubation with AEA. (**B**) CFUs per mL of bacteria treated with 25 µg/mL AEA after various incubation times. (**C**) The pH of *S. mutans* culture medium measured at various incubation times with increasing concentrations of AEA (0–50 µg/mL). (**D**) A time course growth curve of *S. mutans* in the presence of increasing doses of AEA (0–50 µg/mL) as measured by OD_595nm_. *n* = 3; * *p* < 0.05, ** *p* < 0.01, *** *p* < 0.001 in comparison to control samples.

**Figure 2 ijms-24-06177-f002:**
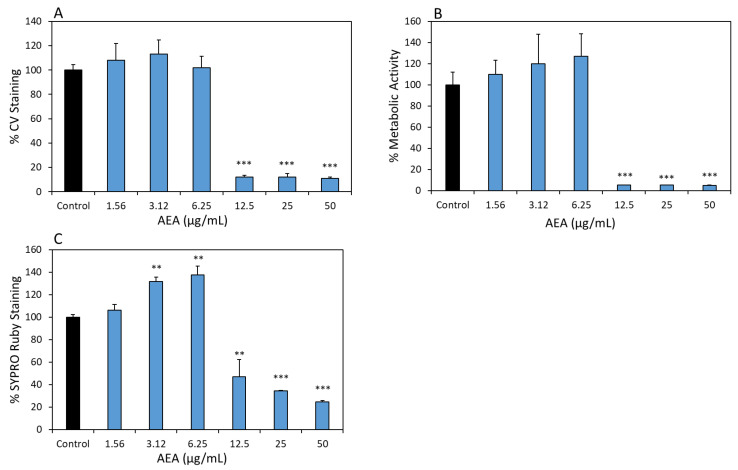
AEA inhibits biofilm formation of *S. mutans*. Biofilm masses of *S. mutans* that had been incubated in the absence or presence of various AEA concentrations or respective ethanol concentrations (0.0156–0.5%) for 24 h. (**A**) CV staining. (**B**) MTT metabolic assay. (**C**) Filmtracer SYPRO Ruby biofilm matrix staining. *n* = 3; ** *p* < 0.01, *** *p* < 0.001 in comparison to control samples.

**Figure 3 ijms-24-06177-f003:**
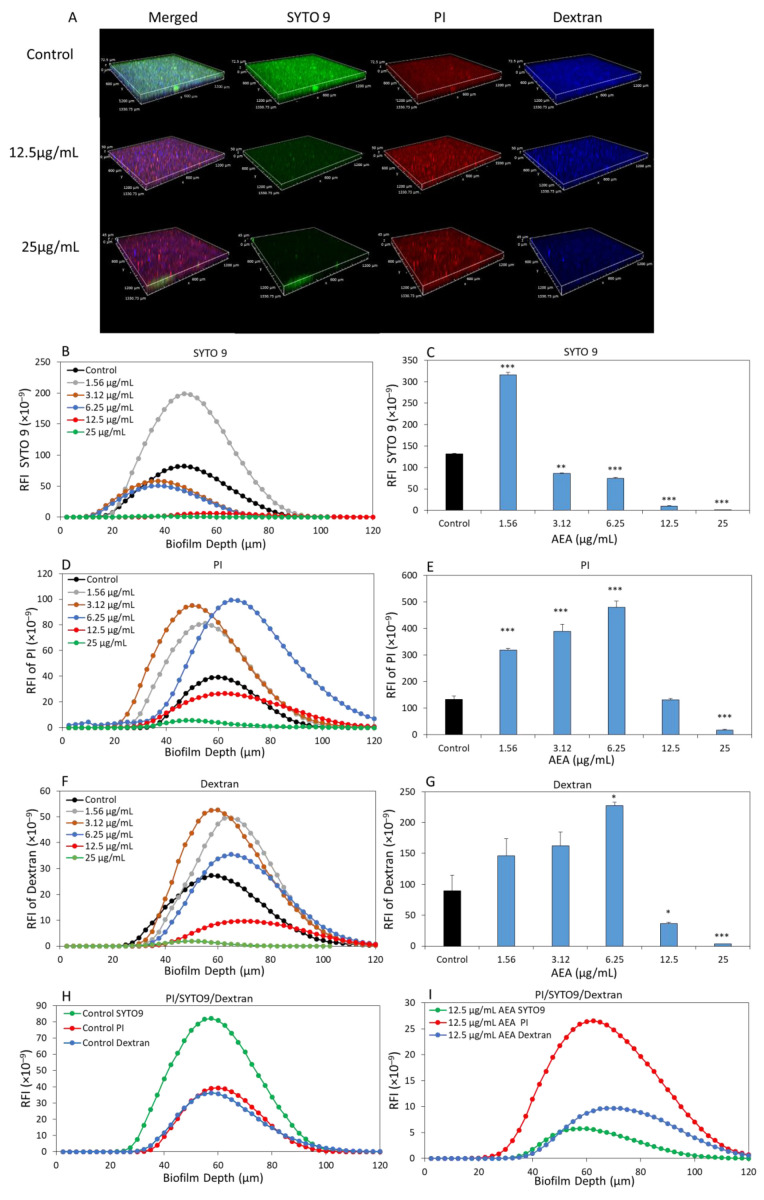
AEA reduces the biofilm mass as well as EPS production of *S. mutans.* (**A**) *S. mutans* biofilms were incubated in the absence or presence of various concentrations of AEA and 0.05% ethanol for 24 h and then stained with Alexafluor^647^ -conjugated anionic Dextran 10,000, SYTO 9 and PI for 30 min. The images were captured by a spinning disk confocal microscope (SCDM). (**B**,**D**,**F**) The relative fluorescence intensities (RFI) of SYTO 9 (**B**), PI (**D**) and Dextran (**F**) in each biofilm layer captured at intervals of 2.5 µm. (**C**,**E**,**G**) Quantifications of the areas under the curves for the three stains. (**H**,**I**) The relative fluorescence intensities of PI/SYTO 9/Dextran in the biofilms of control (**H**) and 12.5 µg/mL AEA-treated samples (**I**). *n* = 3; * *p* < 0.05, ** *p* < 0.01, *** *p* < 0.001.

**Figure 4 ijms-24-06177-f004:**
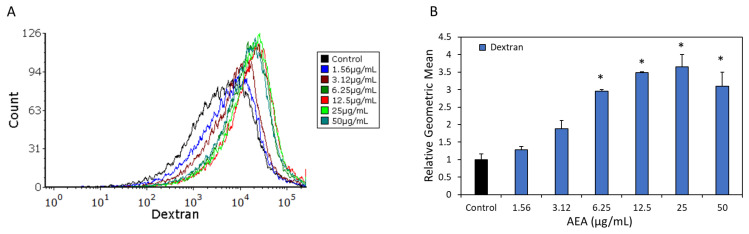
AEA-treated *S. mutans* bacteria exhibit increased affinity to dextran. (**A**) *S. mutans* were incubated in the absence or presence of various concentrations of AEA (1.56–50 µg/mL) or 0.05% ethanol for 2 h and then stained with Alexafluor^647^ -conjugated anionic Dextran 10,000 for 30 min. The fluorescence intensities of the stained bacteria were analyzed by flow cytometry. (**B**) The relative geometric means of the samples presented in (**A**). A total of 50,000 events were collected for each sample. * *p* < 0.05 in comparison to control.

**Figure 5 ijms-24-06177-f005:**
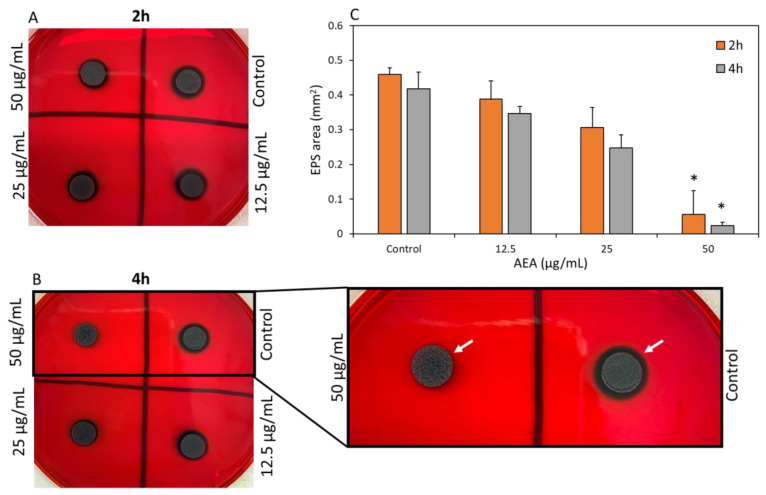
The Congo Red assay shows reduced EPS secretion following AEA treatment. (**A**,**B**) *S. mutans* cultures were incubated in the absence or presence of various concentrations of AEA for 2 h (**A**) and 4 h (**B**), and then 10 µL of each triplicate were plated on Congo Red agar plates containing 1% sucrose followed by a 24 h incubation. The arrows point to the secreted EPS that has turned black in the presence of Congo red. (**C**) Quantification of the black areas around the colonies which represent the secreted EPS. It should be noted that during the short incubation time with AEA, the bacterial growth of AEA-treated bacteria recovered on the Congo Red plates. *n* = 3; * *p* < 0.05.

**Figure 6 ijms-24-06177-f006:**
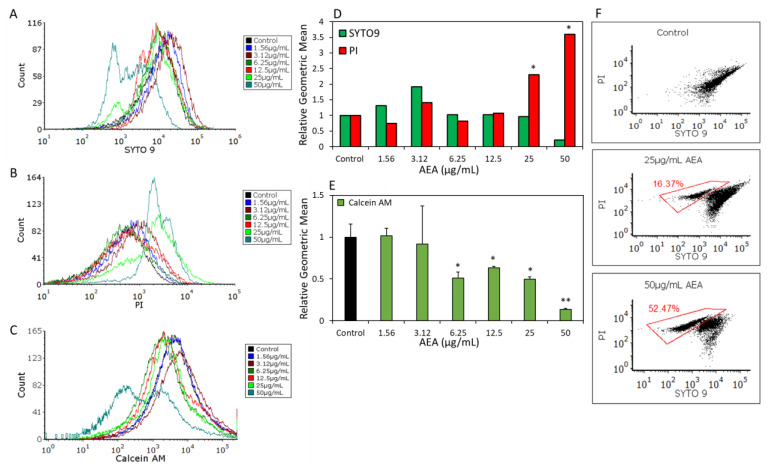
AEA alters membrane properties. (**A**–**C**) *S. mutans* were treated with increasing concentrations (0–50 µg/mL) of AEA for 2 h, and then stained with the nucleic acid dyes SYTO 9 (**A**) and PI (**B**) or with calcein AM (**C**) and analyzed by flow cytometry. (**D**,**E**) The geometric mean fluorescence intensities of SYTO 9 (**D**), PI (**D**) and calcein AM (**E**). (**F**) Dot plots of SYTO 9 and PI samples treated with 25 and 50 µg/mL AEA. The SYTO 9^low^PI^high^ bacterial population is shown in the red gatings. A total of 50,000 events were collected for each sample. * *p* < 0.05 and ** *p* < 0.01 in comparison to control.

**Figure 7 ijms-24-06177-f007:**
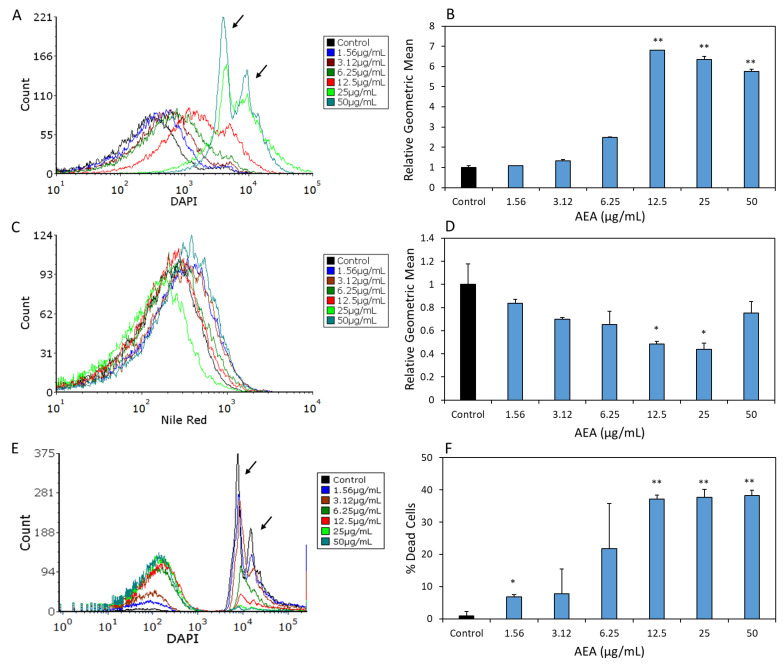
AEA leads to accumulation of DAPI, which might be due to inhibition of the membrane transport system. (**A**,**C**) *S. mutans* were treated with increasing concentrations (0–50 µg/mL) of AEA for 2 h, and then stained with DAPI (**A**) and Nile Red (**C**). (**B**,**D**) The geometric mean fluorescence intensities of DAPI (**B**) and Nile Red (**D**). (**E**) *S. mutans* were treated with increasing concentrations (0–50 µg/mL) of AEA for 2 h, fixed in methanol and then stained with DAPI. The two peaks of the DAPI^high^ population (peak fluorescence intensities of 6786 ± 438 and 13568 ± 345, respectively) reflect the DNA content (1N and 2N) of the cell population (shown by an arrow). The DAPI^low^ population (fluorescence intensity of 10–1000) represents dead bacteria with no or reduced amounts of DNA. (**F**) The percentage of dead cells (the DAPI^low^ population) as measured by flow cytometry. A total of 50,000 events were collected for each sample. *n* = 3; * *p* < 0.05, ** *p* < 0.01.

**Figure 8 ijms-24-06177-f008:**
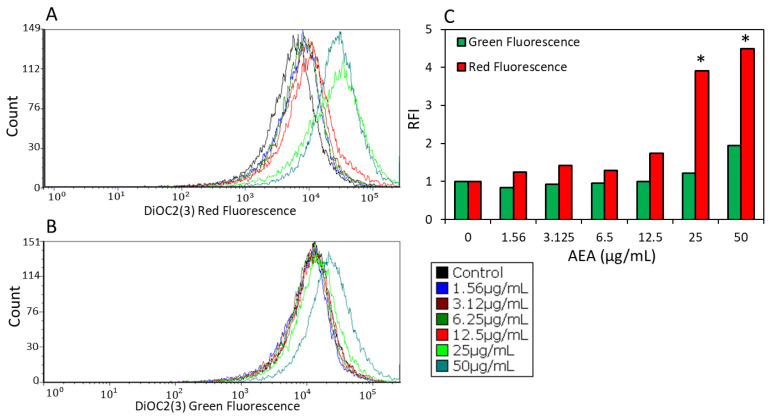
AEA induces immediate membrane hyperpolarization. (**A**,**B**) *S. mutans* were exposed to increasing concentrations (0–50 µg/mL) of AEA followed by DiOC2(3) staining for 30 min and analysis by flow cytometry for red fluorescence (**A**) and green fluorescence (**B**). (**C**) The relative green and red fluorescence intensities (RFI) as measured by the geometric mean of each sample. The RFI intensity was set to 1 for control bacteria. A total of 50,000 events were collected for each sample. * *p* < 0.05.

**Figure 9 ijms-24-06177-f009:**
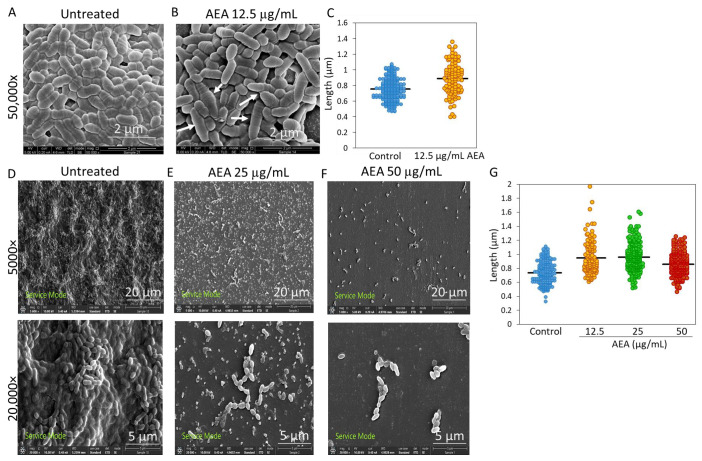
AEA alters the cell length and morphology of *S. mutans*. (**A**,**B**) HR-SEM images (×50,000) of control bacteria (**A**) and bacteria treated with AEA (12.5 µg/mL) (**B**) for 2 h in planktonic growth conditions. The white arrows point to dysmorphic, swollen and/or elongated bacteria. (**C**) The lengths of the planktonic growing bacteria. *n*= 200–220. The black lines represent the average lengths. (**D**–**F**) HR-SEM images (×5000 and ×20,000) of biofilms formed over 24 h in the absence (**D**) or presence of 25 µg/mL (**E**) or 50 µg/mL (**F**) AEA. (**G**) The lengths of the biofilm growing bacteria in the control, 12.5, 25 and 50 µg/mL AEA-treated samples. *n*= 120–220. The black lines represent the average lengths. The images are representative of three independent experiments.

**Figure 10 ijms-24-06177-f010:**
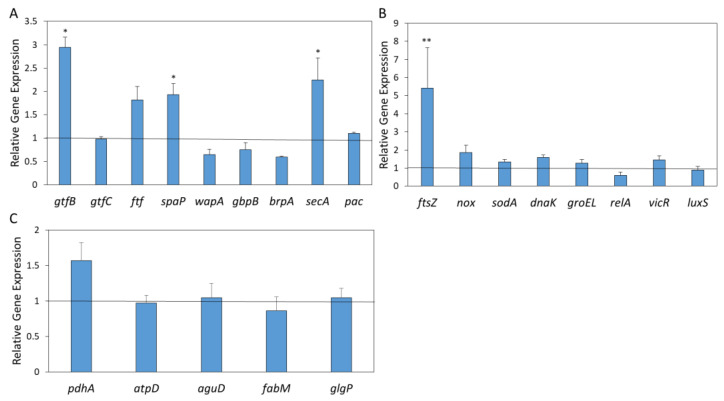
AEA significantly upregulates the gene expression of the cell division gene *ftsZ* in *S. mutans*. *S. mutans* were treated with 12.5 µg/mL AEA for 4 h in BHIS, and the relative gene expressions were determined versus control bacteria grown under the same growth conditions by real-time qPCR using the housekeeping gene *gyrA* as internal standard. (**A**) Genes associated with biofilm formation. (**B**) Genes associated with cell division (*ftsZ*), antioxidant defense (*nox* and *sodA*), stress and stringent responses (*dnaK*, *groEL* and *relA*), and quorum sensing (*vicR* and *luxS*). (**C**) Genes involved in acid tolerance. The experiment was performed in triplicate. The black lines represent gene expression in the control bacteria, which was set to 1. Only genes that were upregulated or downregulated more than two-fold were considered significant. * *p* < 0.05, ** *p* < 0.01 in comparison to control.

**Figure 11 ijms-24-06177-f011:**
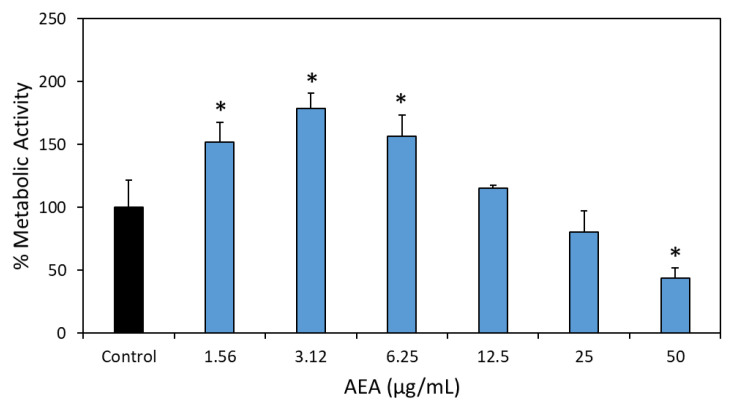
AEA reduces the metabolic activity of preformed biofilms at 50 μg/mL. S. mutans biofilms were allowed to form by growing the bacteria in BHI with 2% sucrose (BHIS) for 4 h prior to treatment with various concentrations (1.56–50 μg/mL) of AEA or respective ethanol concentrations (0.0156–0.5%) for 24 h. *n* = 3; * *p* < 0.05 in comparison to control samples.

**Figure 12 ijms-24-06177-f012:**
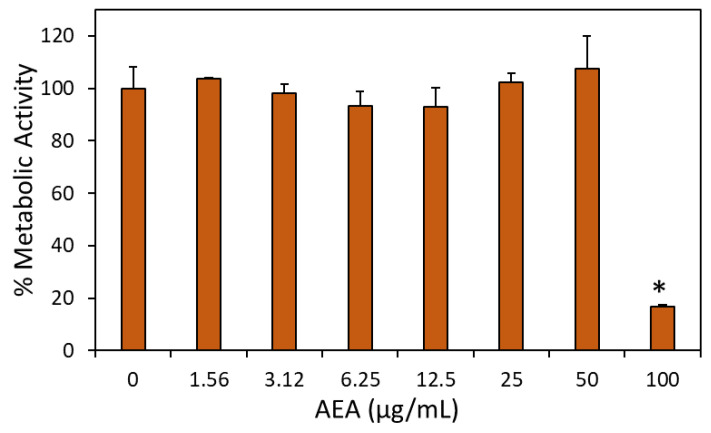
The cytotoxic concentration of AEA on Vero kidney epithelial cells was above the MIC for *S. mutans.* Cytotoxicity assay on Vero kidney epithelial cells was performed by exposing confluent Vero cells to various concentrations of AEA for 24 h. Thereafter, the metabolic activity was measured by the MTT metabolic assay. ** p* < 0.01 in comparison to control cells.

**Table 1 ijms-24-06177-t001:** The minimum inhibitory concentrations (MIC) and minimum biofilm inhibitory concentrations (MBIC) of AEA against the specified oral bacteria.

Species	MIC	MBIC
*S. mutans*	12.5 µg/mL	12.5 µg/mL
*S. sobrinus*	12.5 µg/mL	12.5 µg/mL
*S. sanguinis*	12.5 µg/mL	12.5 µg/mL

**Table 2 ijms-24-06177-t002:** Genes used in mRNA gene expression of *S. mutans*.

Genes	Gene Function	Forward Primers	Reverse Primers	Reference
*gyrA*	DNA gyrase subunit, used as housekeeping gene [45]	TACAGGTGATGTCATGGGTAAATAC	CCGGGTAGTACTTCCATTAGGTCAC	[46]
*gtfB*	Glycosyltransferase-I catalyzes the formation of water-insoluble adherent glucans (one type of EPS) [45]	AGCAATGCAGCCAATCTACAAAT	ACGAACTTTGCCGTTATTGTCA	[21]
*gtfC*	Glycosyltransferase-SI catalyzes the formation of water-insoluble adherent glucans (one type of EPS) [45]	GGTTTAACGTCAAAATTAGCTGTATT	CTCAACCAACCGCCACTGTT	[21]
*gbpB*	Glucan-binding protein B. It mediates cell surface interactions with glucans [45]	AGGGCAATGTACTTGGGGTG	TTTGGCCACCTTGAACACCT	[21]
*brpA*	Biofilm-regulating protein A. It regulates biofilm formation, autolysis, cell division, and acid and oxidative stress responses [45]	GGAGGAGCTGCATCAGGATTC	AACTCCAGCACATCCAGCAAG	[21]
*ftf*	Fructosyltransferase (Levansucrase) catalyzes the formation of fructans (one type of EPS) [47]	AAATATGAAGGCGGCTACAACG	CTTCACCAGTCTTAGCATCCTGAA	[21]
*secA*	Protein translocase subunit SecA involved in the secretion of virulence factors [31]	ATCATGGTACGTGTCACATCAA	CAGAATAATCCTATTGTTGAAT	[31]
*pac*	Major cell surface protein antigen c (PAc) is an adhesin that promotes bacterial adhesion to tooth surfaces [31]	AGCTGGAGAGACAAATGGTTCAT	GACACCAGCAGACTTAGCATCTT	[31]
*dnaK*	A chaperone protein (Hsp70) involved in environmental stress responses [48]	GCAGGTCAAGAGGGAGCTCA	CCGCCCTTGTCTGAGAATC	[21]
*groEL*	A chaperone protein (Hsp60) involved in environmental stress responses [48]	CCAGGAGCTTTGACTGCGAC	TTGCGGATGATGATGTAGATGGT	[21]
*wapA*	Wall-associated protein assists in adhesion to tooth surface and is involved in biofilm formation [49]	GCACGCTTGCAGTACATTGC	CATAAGGTCGCGAGCAGCT	[21]
*spaP*	Surface protein antigen P is cleaved into the two cell surface antigens I/II involved in sucrose-independent adhesion [45]	GACTTTGGTAATGGTTATGCATCAA	TTTGTATCAGCCGGATCAAGTG	[50]
*vicR*	A global response regulator which is part of the two-component VicRK system. It regulates expression of genes associated with cell wall biogenesis and biofilm formation [51]	CGCAGTGGCTGAGGAAAATG	ACCTGTGTGTGTCGCTAAGTGATG	[21]
*nox*	NADH oxidase reduces diatomic oxygen to water when oxidizing NADH to NAD^+^, thereby acting as a reactive oxygen species scavenger [52]	GGGTTGTGGAATGGCACTTTGG	CAATGGCTGTCACTGGCGATTC	[21]
*sodA*	Superoxidase dismutase converts superoxide anions to molecular oxygen and water and thus constitutes a major defense mechanism against oxidative stress [53]	GCAGTGCTAAGACTCCCGAATC	TTGCGGAAGTGTGAGATTGGC	[21]
*relA*	GTP diphosphokinase is involved in the production of (p)ppGpp, which mediates the stringent responses. It regulates biofilm formation and glucose uptake [54]	ACAAAAAGGGTATCGTCCGTACAT	AATCACGCTTGGTATTGCTAATTG	[21]
*ftsZ*	A GTPase cell division protein involved in formation of the Z-ring of the septum [55]	CAACCAAGAGCACAACAGCAAG	ACGACGAAGATTCCAATCGCC	[56]
*luxS*	LuxS encodes for a precursor of autoinducer-2, which is involved in quorum sensing and biofilm formation [57]	ACTGTTCCCCTTTTGGCTGTC	AACTTGCTTTGATGACTGTGGC	[21]
*pdhA*	Pyruvate dehydrogenase A (Acetoin dehydrogenase) is involved in glycolysis. It contributes to bacterial survival in acidic environments [58]	ATGCCAAACTATAAAGATTTAC	TCTTGGGCTTCAATATCT	[59]
*atpD*	The ATP synthase β-subunit of F_1_F_0_-H/ATPase, which produces ATP from ADP in the presence of a proton gradient across the membrane. It promotes tolerance to acidic stress by pumping out protons [60]	CGTGCTCTCTCGCCTGAAATAG	ACTCACGATAACGCTGCAAGAC	[61]
*aguD*	Agmatine:putrescine antiporter comprising the agmatine deiminase system, which produce the alkaline conditions required for acid tolerance [60]	ATCCCGTGAGTGATAGTATTTG	CAAGCCACCAACAAGTAAGG	[61]
*fabM*	Trans-2-decenoyl-(acyl-carrier-protein) isomerase is responsible for the synthesis of monounsaturated fatty acids and is required for survival at low pH (acid tolerance) [62]	ACTGATTAATGCCAATGGGAAAGTC	TGCGAACAAGAGATTGTACATCATC	[63]
*glgP*	Alpha-1,4 glucan phosphorylase catalyzes the rate-limiting step in glycogen catabolism and is thus involved in glucose homeostasis [64]	GACTTTAAAGACACTCTGCATGAAG	ACGAACAACCTTAGCCAAAGAAG	[63]

## Data Availability

Raw data for the figures are available upon reasonable request from the corresponding author.

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
