# Peer review of "Anti-Bacterial and Anti-Biofilm Activities of Anandamide against the Cariogenic Streptococcus mutans"

_ijms, 2023, doi:10.3390/ijms24076177_

Round 1

Reviewer 1 Report

Dear author

In this manuscript, you discussed the effect of AEA on Streptococcus mutans. The effects on the growth of acid-producing biofilm and the morphology of bacteria were studied, and the mechanism was discussed. Although not very in-depth, but relatively comprehensive, and the workload is quite large. In view of this, I have the following questions.

1.       In the PH measure part, you used 50 µg/mL AEA, but in figure 1C Lack the result of 50 µg/mL AEA.

2.       AEA inhibited the total amount of EPS, because the number of bacteria decreased, and related to its inhibitory effect on bacterial growth, did it inhibit EPS produced by individual bacteria?

3.       Figure 5E Is it Calcein AM? Missing annotation. Why is the concentration not exactly the same as PI/SYTO 9.

4.       AEA affects the cell length and morphology of S. mutans. What effect do these effects have on the function of S.mutans? I hope the author adds corresponding content in the discussion section.

5.       Although AEA has shown a relatively good effect in vitro, I am very curious about its specific effect and hope that the author can verify the effect of the caries model as soon as possible.

Author Response

We thank the Reviewer for reading and critically reviewing our manuscript.

In this manuscript, you discussed the effect of AEA on Streptococcus mutans. The effects on the growth of acid-producing biofilm and the morphology of bacteria were studied, and the mechanism was discussed. Although not very in-depth, but relatively comprehensive, and the workload is quite large. In view of this, I have the following questions.

  1. In the PH measure part, you used 50 µg/mL AEA, but in figure 1C Lack the result of 50 µg/mL AEA.

We thank the reviewer for noticing that 50 µg/mL AEA was missed out in Figure 1C. We have now added the data to the graph. 

  1. AEA inhibited the total amount of EPS, because the number of bacteria decreased, and related to its inhibitory effect on bacterial growth, did it inhibit EPS produced by individual bacteria?

Thank you for the comment. In the confocal microscopy data of the biofilms, the reduced EPS observed in the AEA-treated biofilms is mainly due to the reduced adherence of bacteria to the surface and not reduced EPS production per bacteria. In order to clarify this in the text, we have compared the fluorescent dextran (EPS) staining of treated and control bacteria to their own SYTO 9 and PI staining (Figures 3H and 3I) to visualize the ratio of EPS to the number of bacteria. By doing this, we observed that there is relatively more EPS produced per bacterium when treated with 12.5 µg/mL AEA than for the control bacteria. In order to further clarify this issue, we have now also performed a flow cytometry analysis of dextran binding to individual bacteria that have been exposed to AEA or respective ethanol concentration for 2 h, and indeed observed that there is a dose-dependent increase in dextran binding upon increasing concentrations of AEA (Figure 4 – which is a new figure in the revised version). Accordingly, we have added the following text to the result section (lines 181-189): “In order to determine whether there is an alteration in EPS production per bacterium, the fluorescence intensity of dextran was plotted alongside the PI and SYTO 9 (Figures 3H and 3I). We observed that the ratio of dextran staining to SYTO 9 staining was significantly higher in the 12.5 µg/mL AEA-treated biofilms (Figure 3I) than in the control biofilms (Figure 3H), indicating that the amount of EPS produced per bacterium in the biofilms formed in the presence of AEA was higher than that of control bacteria. The reason for this increase might be the release of EPS-producing enzymes upon AEA-induced cell death, which continue to produce EPS.”

And we have added Sub-section 2.4 in Lines 190-197:

2.4. AEA leads to an increased dextran binding to the bacteria

To further study the ratio of EPS production per bacterium, S. mutans was exposed to various concentrations of AEA for 2 h, and then stained with fluorescent dextran followed by flow cytometry. There was a dose-dependent increase in dextran staining upon increasing concentrations of AEA, reaching a maximum at 25 µg/mL (Figure 4A-B). This observation goes along with the SDCM findings (Figure 3H-I). We cannot exclude the possibility that dextran also binds to other components of the bacterial cell surface besides EPS.“  

  1. Figure 5E Is it Calcein AM? Missing annotation. Why is the concentration not exactly the same as PI/SYTO 9.

Thank you for the comment. We have accordingly added the label “Calcein AM” to Figure 6E (previous Figure 5E) and repeated the experiments with all the concentrations (Figures 6C and E).

  1. AEA affects the cell length and morphology of S. mutans. What effect do these effects have on the function of S.mutans? I hope the author adds corresponding content in the discussion section.

The changes caused by AEA on the cell length and morphology of the bacteria may explain the bacteriostatic and bacteriocidal effects of AEA. The elongation of the bacteria suggests that there is a cell division defect. The swelling is a sign that the bacteria have undergone osmotic stress and are dying. The following text has been added to the discussion section to clarify this issue (lines 470-475): “Both S. mutans and S. aureus responded to AEA by becoming enlarged [11], but the mechanism seems to differ. In S. mutans the bacteria become swollen and are on their way to dying, while in S. aureus the enlarged cells had a defective septum arresting them at the last stage of cell division. The swelling of the S. mutans in response to AEA might be a result of osmotic stress due to altered membrane permeability and membrane potential.”

  1. Although AEA has shown a relatively good effect in vitro, I am very curious about its specific effect and hope that the author can verify the effect of the caries model as soon as possible.

It was important for us to first establish the effect of AEA in an in vitro model, but it would be important to study in the future the effect of AEA in a caries model.

Reviewer 2 Report

The authors' group has published papers with similar experimental systems using different materials, which is not novel. If you want to test and compare many materials, you should report them in a single comparative paper.
The experiments also use many experimental models. However, the conclusion is that the antimicrobial activity of AEA is bactericidal and the amount of EPS is relatively reduced accordingly, which is not topical.
The mechanism of action, biological effects (protein and gene level), etc. have not been studied.
The results of this paper elucidate that AEA is effective against S. mutans floating bacteria, but there is a lack of experiments to explain the anti-biofilm effect.
Biofilms have been cultured in the presence of AEA, and the penetration and bactericidal effect on mature biofilms remains unknown.
In the oral cavity, saliva and swallowing dilute the concentration of AEA and cannot hold more than the MIC.

Author Response

We thank the Reviewer for reading and critically reviewing our manuscript.

The authors' group has published papers with similar experimental systems using different materials, which is not novel. If you want to test and compare many materials, you should report them in a single comparative paper.

Our biofilm laboratory focuses on compounds with anti-bacterial and anti-biofilm activities, and we have studied their effects on various bacteria and fungi. From a clinical aspect, it is important to study the spectrum of responsive species. It is well known that each species responds differently to the different compounds, and it is known that their responses and sensitivities might vary. Concerning AEA, we have previously focused our study on drug-resistant Staphylococcus aureus species where we revealed several action mechanisms including efflux pump inhibition with sensitization to antibiotics. However, so far, a comprehensive study on its action on Streptococcus mutans has not been performed, and therefore, this study was conducted. In our discussion, we have in some places compared the mode of AEA action on S. mutans with those observed in previous studies for S. aureus. We found both similarities and differences, pointing to species-specific actions, which are important to specify. Therefore, it is important to document each system separately in order to elaborate in-depth on the individual responses. The idea of combining all of them into one paper would fit into a future review article. We have rewritten some of the text in the discussion section to better emphasize the AEA effect on S. mutans.

The experiments also use many experimental models. However, the conclusion is that the antimicrobial activity of AEA is bactericidal and the amount of EPS is relatively reduced accordingly, which is not topical.

This is the first study to demonstrate the anti-bacterial and anti-biofilm effect of AEA on S. mutans using different approaches. The experimental models used in our study helped to investigate the mode of action AEA has on S. mutans and we are planning in our future studies to investigate AEA in a caries model. Additionally, each method provides a different aspect that all together provides a solid basis for the conclusions.

The mechanism of action, biological effects (protein and gene level), etc. have not been studied.
The results of this paper elucidate that AEA is effective against S. mutans floating bacteria, but there is a lack of experiments to explain the anti-biofilm effect.

The study has provided data on some action mechanisms including interference with cell division, membrane hyperpolarization, alterations of membrane permeability, bactericidal effect as well as anti-biofilm effect. The few bacteria that attached to the surface in the presence of AEA appeared dead, such that some of the anti-biofilm effects are due to the bactericidal action. Another mechanism for the anti-biofilm effect is reduced EPS production, which is due to fewer adherent bacteria. We have now performed gene expression studies to understand the mechanism behind the anti-biofilm effect, but the observed alterations in gene expression of biofilm-associated genes couldn't explain the anti-biofilm action mechanism. We have nevertheless added these data to the revised version of the manuscript (Figure 10A). Importantly, we found that the cell division gene ftsZ is significantly upregulated by AEA (Figure 10B). We also studied the effect of AEA on gene expression of genes involved in acid tolerance (Figure 10C), however, no significant changes were observed. As discussed in the text, it is likely that AEA acts at the post-translational level. One indication for this is the immediate hyperpolarization caused by AEA, as well as the intracellular accumulation of DAPI indicative of inhibition of membrane transport. As the sequela leading to EPS production requires membrane transport, this might be one mechanism for the anti-biofilm effect, as demonstrated by the Congo red assay.

Biofilms have been cultured in the presence of AEA, and the penetration and bactericidal effect on mature biofilms remains unknown.

We thank the reviewer for this comment. It is important to be able to eradicate preformed biofilms. We have now added data showing partial effect of AEA on preformed biofilms (Section 2.11 and Figure 11), although higher concentrations (50 µg/mL AEA) were required to reduce the preformed biofilms than the MBIC of 12.5 µg/mL necessary for the prevention of biofilm formation. Further studies should be pursued to look for combined drug treatment that can increase the efficacies toward mature biofilms.

In the oral cavity, saliva and swallowing dilute the concentration of AEA and cannot hold more than the MIC.

This is a good point, and further studies should address this issue using a constant depth film fermenter (CDFF) which would mimic the conditions in the oral cavity in vivo.

Reviewer 3 Report

Very interesting work, it subscribes to the current trends on the use of cannabinoids and derivatives in medicine and preventive treatment. I have no major comments except:

1. text editing: number and unit should not be on separate lines (line 99 and 101)

2. point 4.1 - ethanol is a toxic for cells, bacteria.... Maybe The effects of the study can be enhanced by the effect of pure ethanol? I know that pure ethanol was taken as a control but there is such a thing as a synergistic effect. Do you checked this?

3. point 4.2 - on the figures on the results part of this publication we have only one control on the graphs but here we have two controls: (1) untreated bacteria;(2) ethanol-treated bacteria. Which one is presenting in the results?

4. For a complete study, what I am missing here is an assessment of AEA's cytotoxicity. Did the authors of the publication check the cytotoxicity of this compound?

Author Response

We thank the Reviewer for reading and critically reviewing our manuscript.

Very interesting work, it subscribes to the current trends on the use of cannabinoids and derivatives in medicine and preventive treatment. I have no major comments except:

  1. text editing: number and unit should not be on separate lines (line 99 and 101)

We are aware that the unit appears in the following line. But, we had to follow the layout according to the IJMS instructions and are therefore unable to change it without deviating from the required journal style.

  1. point 4.1 - ethanol is a toxic for cells, bacteria.... Maybe The effects of the study can be enhanced by the effect of pure ethanol? I know that pure ethanol was taken as a control but there is such a thing as a synergistic effect. Do you checked this?

AEA was dissolved in pure ethanol at 10 mg/mL, but for the experiments it was diluted 1:200 for the highest tested concentration of 50 µg/mL AEA, resulting in the highest ethanol concentration of 0.5% in our experiments. Thereafter serial two-fold dilutions were made, with respective lower concentrations of ethanol. The diluted ethanol didn’t have any effect on the bacterial growth, so all the effect observed comes from AEA. But the ethanol controls had to be added to all of the experiments as it is a vehicle control. Pure ethanol causes fixation of the bacteria and all tissues, and can thus not be used in its pure state (and can induce “a drunken state”). In order to better clarify this issue we have added the % of ethanol concentrations used to the methods section (lines 531 and 541).

  1. point 4.2 - on the figures on the results part of this publication we have only one control on the graphs but here we have two controls: (1) untreated bacteria;(2) ethanol-treated bacteria. Which one is presenting in the results?

All experiments were performed with two controls: untreated bacteria and bacteria treated with ethanol at concentrations of 0.0156%- 0.5%. Ethanol was used as a control because AEA was dissolved in ethanol. Since the presence of the diluted ethanol had no effect on the bacteria as compared to the untreated bacteria, all the data presented show results compared to the untreated bacteria. To clarify this issue we have added the following text to the method section “Respective dilutions of ethanol (0.0156% - 0.5%) as well as untreated bacteria were used as controls in this study.” (lines 530-532).

  1. For a complete study, what I am missing here is an assessment of AEA's cytotoxicity. Did the authors of the publication check the cytotoxicity of this compound?

We thank the Reviewer for the question. We have now added data concerning the cytotoxicity of AEA on normal Vero epithelial cells, showing no cytotoxicity at the concentrations used on S. mutans (1.56-50 µg/mL). 100 µg/mL AEA was cytotoxic to these cells. These data are now presented in Figure 12 and Subsection 2.12 added to the Result section. This issue has also been addressed in the discussion.  It should be mentioned that AEA is a natural compound produced by our body, and it has been used in vivo in mouse models of inflammation.

Round 2

Reviewer 2 Report

The experiment is just AEA (endocannabinoid anandamide) on biofilms of S. mutans and S. mutans infants.
In the revise test, the only analysis of the mechanism of action of the effect, etc. (huge amount of data on the relevant genes) is added.

However, the previous biofilm experiment remains poor.
Only Figs. 2, 3, and 9 verify the anti-biofilm effect, which are mixed and static (infant model) 24h cultures.
No substances were acted upon after biofilm formation.
No experimental systems for mature biofilms were performed.
This point has not been improved. This study does not constitute a screening study for clinical application.

The authors' group has published similar papers with different materials.
As it stands, it is unclear which material the authors are recommending.
If the authors are considering clinical applications in the future, they should disclose all their data at once, including comparisons with other materials.
We cannot gauge the authors' strategy for cariogenic biofilms.
It should be indicated in the introduction section.

Author Response

We thank the reviewer for critically reading our manuscript.

The experiment is just AEA (endocannabinoid anandamide) on biofilms of S. mutans and S. mutans infants.

In the revise test, the only analysis of the mechanism of action of the effect, etc. (huge amount of data on the relevant genes) is added.

However, the previous biofilm experiment remains poor.

Only Figs. 2, 3, and 9 verify the anti-biofilm effect, which are mixed and static (infant model) 24h cultures.

No substances were acted upon after biofilm formation.

No experimental systems for mature biofilms were performed.

This point has not been improved.

We indeed added data on the effect of anandamide on pre-formed biofilms (Figure 11) as requested by the Reviewer in round #1. We are aware that anandamide is more effective in preventing biofilm formation than reducing preformed biofilms, which is also observed for many other anti-biofilm agents. We have added the following text to clarify this issue: “AEA was found to be more efficient in preventing biofilm formation than to act on preformed biofilms, although a certain reduction in preformed biofilm could be observed with the higher concentration of 50 µg/mL AEA. This is actually not surprising as AEA is a hydrophobic molecule, while the EPS of the biofilm is relatively hydrophilic and thus is expected to act as a drug barrier. Further studies are required to find the drug combination that can overcome this drug barrier, thus enabling the action of AEA and other anti-bacterial drugs on mature biofilms.”

This study does not constitute a screening study for clinical application.

We agree with the reviewer that our study is not a screening study. The focus of our study was to understand the anti-bacterial and anti-biofilm activities of anandamide on the cariogenic S. mutans bacterium. It is important to manifest the anti-bacterial and anti-biofilm activities of anandamide prior to proceeding to clinical studies.

The authors' group has published similar papers with different materials.

As it stands, it is unclear which material the authors are recommending.

The current manuscript documents the anti-bacterial and anti-biofilm activities of anandamide against a cariogenic bacterium, which have not been reported before. Further studies are required to test its application in clinical settings.

If the authors are considering clinical applications in the future, they should disclose all their data at once, including comparisons with other materials.

A comprehensive comparison between the different known anti-biofilm compounds belongs to a review article. In the discussion we have done a comparison with other agents and other bacteria.

We cannot gauge the authors' strategy for cariogenic biofilms.

It should be indicated in the introduction section.

Our study is a basic research intended to understand the action mechanism of anandamide. Further studies are required to find the right drug combination that can eradicate pre-formed biofilms of cariogenic bacteria. To clarify this issue, we have modified the last sentence of the Introduction: “The aim of this study was to investigate the anti-bacterial and anti-biofilm effects of AEA on cariogenic S. mutans and shed light on the action mechanism.”

Round 3

Reviewer 2 Report

Comments from reviewer,

The authors have determined that AEA has only an inhibitory effect on biofilm formation, based on data on genes expressed by AEA on biofilms.
And only static system models have been used to conduct the experiments.
They do not conduct it contrary to the reviewer's instructions.
Phenomenologically, the effect on biofilm formation remains unknown.
No experiments have been conducted to confirm this.
In general, to search for anti-biofilm effects, it is common to screen with a static system model and then conduct a mature biofilm model. (no matter what the substance).
We believe that confirmation work as a phenomenology is necessary.
FOR EXAMPLE:
Maezono H et al: AAC 2011, 55: 5887-5892.
for a paper on this topic.
Two of the five antimicrobial agents were effective against adherent cells, but one of the two that were effective against adherent cells was effective against the biofilm that formed.

Other.
Let's assume, that AEA only inhibits cell adhesion or biofilm formation.
Do you believe that targeting only S. mutans in cariesogenic biofilms will reduce caries?
Do you believe that AEA can be applied clinically?
Please consider the significance of antimicrobial experiments using static biofilm models of single bacterial species in vitro when targeting oral biofilm infections.
Dental biofilms are genetically inhabited by 700-800 species of bacteria.
This reviewer believes that the term caries-associated biofilm (bacterial species) should be used strictly instead of caries-pathogenic biofilm (bacteria).
It seems to me that we need to develop substances that nonspecifically destroy biofilm bacteria or act on the entire biofilm structure.
As further discussion would be unconstructive, I leave the conclusion to the editor.

No further peer review will be done.
